# Parent and Clinician Perspectives on the Participation of Children with Cerebral Palsy in Community-Based Football: A Qualitative Exploration in a Regional Setting

**DOI:** 10.3390/ijerph17031102

**Published:** 2020-02-10

**Authors:** Carmel Sivaratnam, Katherine Howells, Nicole Stefanac, Kelly Reynolds, Nicole Rinehart

**Affiliations:** 1Deakin Child Study Centre, School of Psychology, Faculty of Health, Deakin University, Geelong 3220, Australia; khowells@deakin.edu.au (K.H.); nicolestefanac@gmail.com (N.S.); nicole.rinehart@deakin.edu.au (N.R.); 2Kids Plus Foundation, P.O. Box 6251, Highton 3216, Australia; Kelly.Reynolds@kidsplus.org.au

**Keywords:** cerebral palsy, physical activity, community participation

## Abstract

The current study aimed to qualitatively explore parent and clinician perspectives on the factors influencing participation in a community-based Australian-Rules Football program for five to 12-year-old children with cerebral palsy (CP) in a regional setting. Six allied-health clinicians and two parents of children with CP participated in focus groups exploring factors influencing participation. Thematic analysis indicated seven key factors influencing participation—of which, six were environmental factors and one was related to child characteristics. Environmental factors included resources, communication, knowledge and previous experience, attitudes and expectations, game factors and community relevance. Child characteristics included age, preferences, confidence, as well and cognitive and physical functioning. Notwithstanding limitations, the current study highlights the central role of ‘people factors’ in the child’s environment, in facilitating participation in community-based physical activity.

## 1. Introduction

Participation in regular physical activity (PA) has consistently been linked to positive outcomes in children of all abilities, spanning physical, social, behavioural, psychological, emotional and cognitive domains [1]. Moreover, participation in community-based PA has been found to provide additional benefits for children with a range of developmental disabilities [2], among which are social skills [3], social connection and a sense of belonging to community [4]. Despite the documented benefits to participation, children with cerebral palsy (CP) participate less frequently in physical and social activities, and in a smaller variety of activities than children without CP [5]. While age and severity of gross motor impairment impact on the type, intensity and frequency of PA participation [6,7] children with CP have overall lower levels of PA participation [8], and higher levels of sedentary activity participation than those without CP [9]. Given the adverse health implications of a sedentary lifestyle [9], there has been an increased focus on understanding the facilitators and barriers to participation in PA in this group.

### 1.1. Frameworks of PA Participation

Aligning with the International Classification of Functioning (ICF), Disability and Health definition of disability, which conceptualises the term disability as an umbrella term covering impairments, activity limitations, and participation restrictions [10], current empirical research suggests that participation in PA is influenced by a convergence of personal and environmental factors [11]. The Family of Participation Related Constructs (fPRC; [12]), highlights the dynamic interplay between child-related and environmental factors, which influence and are influenced by activity participation, attendance and involvement. This framework purports that an understanding of environmental and social barriers is imperative in providing a holistic conceptualisation of participation in PA [13]. Environmental factors, a core component of the ICF Framework, have been found to pertain to social and familial attitudes and relationships, the physical facilities within and providing access to a setting, as well as policy and organisational characteristics [14,15,16].

As illustrated within the ICF Framework, the concept of personal factors influencing PA participation in children include internal characteristics such as age and gender, preferences for physical activities, and previous participation in PA [13]. There is overlap between psychological factors and body function within this framework, with factors such as the child’s level of mastery motivation and perceived competence falling within the ICF notion of Body structure and function. The presence of intellectual disability has also been found to influence participation in PA, where cognitive impairments are seen to impact a number of cognitively-based personal factors facilitating participation, which include planned behaviour [17] and self-determination [18].

Biological and physical factors relating to the child’s primary or co-morbid conditions (e.g., congenital heart abnormalities or obesity [19]) are also seen as salient factors influencing participation. These characteristics are predominantly conceptualised as non-modifiable factors [16] and come under the ICF concept of Body structure and function. The child’s activity limitations, including the severity of behavioural and social difficulties as well as adaptive functioning skills, specifically, the ability to carry out daily living skills, have also been found to impact physical and leisure activity-participation in a number of studies [20,21].

### 1.2. Factors Influencing Participation: Current Knowledge and Gaps in Knowledge

Aligning with factors within the ICF framework, studies exploring PA participation have implicated personal, social, as well as environmental, policy and program-related facilitators and barriers for children and adolescents with CP [14,22]. There has a been a primary focus within the literature on exploring the factors influencing PA broadly in CP populations [23]. However, more recent research has begun to explore the barriers and facilitators which exist to community-based PA participation specifically, where a qualitative study involving adolescents with CP concluded that tailored activities, proactive coaching styles and the desire for health and fitness are primary facilitators of participation [24]. To further this, studies on the participation patterns of children with CP suggest that participation in this group is influenced by geographical location [25,26], where participation is higher in regions which enable greater environmental accessibility [27,28]. It is therefore unsurprising that a recent US study indicated that individuals with CP living in rural areas experience additional barriers to community-based PA (e.g., accessibility to adequate equipment/environmental structures, isolation) [29]. To the best of our knowledge, however, no study has yet examined such barriers and facilitators within regional and rural areas among Australian communities.

Furthermore, while emerging research is beginning to incorporate the perspectives of clinicians (who interact with children on a regular basis) on the factors influencing community PA participation in children with CP [22,24], there remains a primary focus on parent and child perspectives. While the perspectives of parents and children are vital, the dearth of perspectives of the broader system involved in the care and management of the child’s health has hindered a holistic view of the factors influencing participation. Moreover, current research suggests that skills-based intervention groups may not translate into activity participation [30], suggesting that there is a gap in knowledge around the applicability of skills learned in clinical settings into participation in real life everyday settings. It is recognised that staff and/or coach attitudes, as well as the provision of support and information by staff has been found to be a key factor influencing participation in community-based activities [14,24,27] and that parent advocacy facilitates participation in populations of children with disability [31]. However, it has also been suggested that clinicians are the ‘missing link’ to community-based PA participation for children with disabilities, given that they are often seen as reputable sources of information in their communities with regards to PA, and possess expert knowledge on the child’s disability [32]. To further this, Palisano et al. [33], when examining participation in physical/occupational therapy for physical disabilities, highlighted the importance of collaboration between the child, their family and clinicians/professionals as a factor in facilitating participation. Without the inclusion of clinician perspectives in research into participation in community-based PA, little is currently known about the distinct and overlapping roles of parents, clinicians and community sporting clubs in influencing the participation of children with CP in these activities.

Taken together, there is a need for increased research into the factors which influence participation in community-based PA programs in regional Australian sporting settings, whilst considering clinician perspectives. The current study, therefore, aimed to qualitatively explore clinician and parent perspectives on the factors influencing participation in a community-based Australian-Rules Football program for primary school-aged children with CP in a regional setting. Given the centrality of football in Australian culture, a lack of understanding of the factors influencing participation has the potential to hinder the inclusion of children with physical disabilities in a community-based activity which is ingrained in Australian culture and identity. Whilst current research has begun to examine factors influencing participation for children with disabilities in this sport [2,3] this research is limited to children with autism spectrum disorder (ASD), or a mixed group of children with developmental disabilities (inclusive of children with CP), and are not specific to a regional setting. To the best of our knowledge, this is the first study to examine these factors in relation to Australian rules football for children with CP specifically within a regional setting. Focus groups were conducted with both parents and clinicians in order to generate rich, in-depth discussion based on different stakeholder perspectives. Thematic analysis of qualitative interviews was undertaken to explore facilitators and barriers to participation.

## 2. Materials and Methods

### 2.1. Participants

#### 2.1.1. Clinician Focus Groups

Six female clinicians took part in two separate clinician focus groups. Four clinicians were qualified physiotherapists, one was an occupational therapist and one was a speech pathologist. Clinicians were recruited from a regional disability service provider that provides intervention for children with a range of physical and developmental challenges. Given that this study aimed to understand clinician perspectives on the participation of children with CP in a regional setting specifically, recruitment of this group targeted one clinical service provider which met all of the above criteria, namely, a specialist CP service provider, staffed by clinicians, and was situated in a regional town. All clinicians within this service were invited to participate given that they all worked with children with CP within their specific disciplines (physiotherapists, occupational therapists, speech pathologists).

#### 2.1.2. Parent Focus Groups

Two parents/carers took part in one parent focus group. Parents were recruited from the local community following dissemination of advertising material through community health services, noticeboards, not-for-profit support organisations, CP-specific research registries and local sporting clubs. Parents/carers were invited to participate if they had, or cared for, a child with additional physical needs such as cerebral palsy. Participants were a mother of an eight-year-old girl with a diagnosis of CP (gross motor function classification system (GMFCS I)) currently participating in community-based football and the father of a six-year-old boy with a combined diagnosis of Autism Spectrum Disorder (ASD) and CP (GMFCS I). This child had participated in a local come-and-try day for a community-based football program but had not signed up to the program at the time of the focus group.

### 2.2. Procedure

Participants were sent the focus group questions (refer to Appendix A) one week prior to the session, to enable participants to ask questions about their participation, opt-out if they wished, as well as to allow time to consider their responses and bring notes if they wished. All focus groups were run at a local community clinic in a regional location in Victoria, Australia and were facilitated by an experienced focus group moderator who was a senior physiotherapist with experience working with children and adults with physical disabilities. A research assistant trained in conducting research with children with disabilities and their carers was also present at the session to assist with audio-recording the session. Participants were provided with the opportunity to ask questions before the commencement of the focus groups and at the end of the session. Sessions ran for approximately 60 min each and were audio-recorded for subsequent transcription. Focus group questions explored facilitators and barriers to participation in the community football activity, as well as the relevance of football participation within the broader regional community for children with physical disabilities. Focus group items were based on the theoretical framework for participation and involvement proposed by Imms et al. [34] for children with disabilities, in relation to environmental (e.g., access to participation) and child specific factors (e.g., preferences) influencing participation. Handwritten notes were also taken during the session to further capture the salient information gleaned in each session. Ethical approval was obtained from Deakin University Human Research Ethics Committee (Project Number # 2016-225) on 9 August 2016, and written informed consent was obtained from all participants.

### 2.3. Data Analysis

Audio-recordings from the focus groups were transcribed verbatim for each session in its entirety and analysed using NVivo software (Version X, QSR International). Thematic analysis was conducted at a semantic level in order to allow for an in-depth exploration of themes arising from the focus groups, as outlined by Braun and Clarke [35]. Two researchers independently read the transcripts several times to familiarise themselves with the data (N.S. and K.H.). Initial codes were then derived line-by-line from the data by both researchers independently to ensure credibility of the data. These codes were not preconceived, andprovided a label for a feature of the data that was considered potentially relevant with as little interpretation as possible. As such, the number of codes were not restricted, and concepts were identified as they emerged. The data of each focus group was analysed separately and then codes were collapsed by participant group (i.e., clinicians and parents) to get an overview of the views of that subgroup.

After coding, similar concepts were grouped together to form overarching themes that captured a unique feature of the data. Researchers (N.S. and K.H.) then engaged in discussions about whether or not a theme represented the views of all participant groups, and rich descriptions were used to exemplify sources. As themes covered similar concepts across both parent and clinician focus groups, these were combined to encompass all reported factors influencing participation regardless of participant group. At this point, original transcripts were also re-read to ensure that no significant aspect of the data had been overlooked. Names were then created for each theme and extracts were drawn from the data to represent each theme.

The original transcripts, together with codes and themes derived by the first two researchers were then reviewed by a third researcher. Through face-to-face discussions, the three researchers resolved any inconsistent interpretations which arose in the analysis of the transcripts and the emerging codes and themes and agreed upon relevant quotations to use in order to convey themes derived from the analysis.

### 2.4. Credibility and Qualitative Rigor

In order to ensure dependability of interpretations [36], the initial identification of codes and themes were carried out by each researcher independently. Through face-to-face discussions, the two researchers resolved any inconsistent interpretations which arose in the analysis of the transcripts and the emerging codes and themes and agreed upon relevant quotations to use in order to illustrate themes from the analysis. The list of themes generated were sent to participants to review. Words and phrases used by participants were incorporated, where possible, in the construction of codes and themes, in order to maximize credibility.

## 3. Results

Qualitative analyses of the focus groups generated seven themes relating to factors influencing participation, namely, child characteristics, resources, communication, knowledge and previous experience, attitudes and expectations, game factors and community and cultural relevance. Figure 1 below outlines the themes derived from the qualitative analyses.

### 3.1. Child Characteristics

Clinicians and parents frequently discussed a child’s physical and cognitive capacity as influencing football participation:
“... You might have a child in a wheelchair, you might have some in a walking frame, and you might have some that can... walk independently… often the speed of sport can be too fast for the kids”—Clinician
“Kids who have CP have such a broad range of cognitive capacities and their communication”—Clinician

In addition to the child’s physical and cognitive functioning, age was also reported to impact upon participation.


*“I think, he’s just that in that age group yet like, where he is, you know six, he’s still not that interested in a lot of things”*
—Father of a six-year-old boy

A clinician mentioned that as a child progressed through primary school, participation in a community-based football program was perceived to be *“…a bit daggy”*.

Children’s sensory impairments arising from co-morbid diagnoses, particularly Autism Spectrum Disorder in this instance, were identified as a barrier to participation.


*“…some of it is the mental bit, “oh the balls wet”, you know “I don’t want to touch it”, things like that”*
—Father of a six-year-old boy

Having a personal interest in or a preference for football over other activities was seen to be a facilitator to participation in community football.

Child confidence was also identified as a barrier to participation by a parent:
“…well for Brad (name anonymised to protect participant identity), I don’t think it’s that he doesn’t know he can’t do it, I think he just hasn’t got that confidence yet…”—Father of a six-year-old boy

Furthermore, sport was seen to highlight physical differences between children with physical disabilities and their peers.


*“… I think it’s almost as if the kids that are more physical able, your GMFCS I’s and high functioning II’s, that notice it. Like they’re the ones that realise their different cause they’re almost there”*
—Clinician

Clinicians also discussed that with age, children increasingly compare themselves to the children around them and notice a difference, with one mentioning that a boy she worked with when playing in a team was *“…really aware that he wasn’t as good as his friends”*.

### 3.2. Resources

Availability of family resources was also identified by clinicians as a factor influencing participation, where time and financial limitations for families were commonly identified by clinicians. Given the high volume of equipment and interventions required by children with a physical disability, the cost involved in football participation was highlighted as an important consideration. Furthermore, clinicians mentioned that parents of children with disabilities often have many other competing commitments.

At the club level, adapting the physical environment was also discussed as a facilitator to participation:
“not that we need to go and adapt and modify everything but for the club to be aware of what works within their environment or what kind of access is possible in terms of...you know the toilet access, or parking, and all those sorts of things....”—Clinician

The same clinician reflected that if the club did not have adapted access/parking available, information should be provided to families before they enrol their child through avenues such as enrolment forms or club websites.

Parents also highlighted the need for clubs to provide children with additional needs opportunities to try the sport before electing to sign up to an entire season.


*“…if they do have a one-off, I know it’s again, it’s like anything, trying to find the day and people to come, but umm they can see that kids of all ages and all levels of disabilities can participate. And if they like it”*
—Father of a six-year-old boy

### 3.3. Communication

Clinicians frequently emphasised the need for a centralised system of communication for parents, coaches and professionals involved in the child’s care. Furthermore, clinicians also discussed the need for a *“key contact”* or *“inclusion officer”* at clubs or through club websites that families could contact and communicate with before starting at the club. One clinician also highlighted that this may facilitate education at a club level.


*“…that might help with the education, wider (repeated) reaching education, if clubs are encouraged to have that inclusion officer and they attend a day of education…”*
—Clinician

Clinicians and parents also emphasised the importance of a “*clear communication channel”* between clinicians and coaches to enable discussions around strategies that support the child. Having clear channels of communication between coaches and clinicians was identified as a way to share knowledge, where clinicians could provide coaches with written summaries of each child’s condition, their strengths and limitations as well as tips on inclusion. It was discussed that coaches could also feedback information to the clinician relating to the sport-specific skills needed, so they could incorporate this into the child’s therapy sessions and tailor a child’s therapy session to a particular sport so they can transfer those skills into an everyday setting. This sharing of knowledge was discussed as being presented in the form of tip sheets, fact sheets as well as practical advice.

A parent also reflected on the importance of communication between the parent, the child and the coach.


*“...instead of me giving Rebecca (name anonymised to protect participant identity) strategies of how to deal with that, but the coach giving her strategies as well. So that she’s getting, you know, the same message but from different people and not just getting it from me and disregarding it…”.*
—Mother of an eight-year-old girl

Clinicians frequently highlighted the need for a summary resource of information pertaining to the child’s abilities and needs to ensure that coaches running the sessions have access to information on strategies to support the child.


*“So if this child comes and these are the additional strategies or things that we’re using that supports that child to be here, that information is consistent and passed on to anybody who’s running the group that day”*
—Clinician

Some parents and clinicians identified that the parent’s ability to advocate for their child’s needs with coaches and clinicians played a role in facilitating the child’s participation.

*“Like you know it comes down to the parents as well and having that confidence and being the advocate…”*.—Clinician


*“…When she starts something new… I just go and speak to the coach and say this is it, you know. Have a go. If you have any issues, let me know. We’ll talk to the physio and we’ll work through... If you’re finding that she can’t do something, talk to me about it and let’s talk about her CP and how it’s affecting that.”*
—Mother of an eight-year-old girl

### 3.4. Knowledge and Previous Experience

Children’s previous exposure to football and more broadly sport was discussed by parents as a facilitator to football participation. One parent of a six-old-boy reflected on how his older son participates in a community-based football program, and his six-year-old son had attended and enjoyed a session as part of a bring-a-friend day. This father also reflected on the fact his son joined in the activities on this day:
“…he was there, and he had fun and you know he said he had fun…”

A family’s current and previous involvement in football was also seen as a factor influencing football participation:
“…the parents aren’t playing, so the kids don’t play and then that whole cycle…”—Clinician

The clubs’ and coaches’ previous knowledge and experience working with children with additional needs was also seen as a factor influencing participation. A clinician discussed that while many clubs would not decline having a child with disability join, they may lack the skills and experience in including children with disabilities:
“but I think that it would get a ‘ohhhh what do we do...’ because I reckon there is a lack of experience…”—Clinician

Coach confidence was also discussed as a significant factor.Specifically, clinicians discussed that that there may be a lack of knowledge or understanding of the condition the child may have, and the risks of injury are for that child in playing.


*“…it’s that concern for the child’s safety or you know, worried they might knocked by someone else or they might get urm so you know that’s genuine concern but having the knowledge to know that’s ok and [repeated], yeah what the risks are or alleviating some of that concern can go a long way to…helping with just general confidence.”*
—Clinician

Clinicians identified that coaches engaging in training around inclusive practices was a facilitator to participation.


*“…even having training for clubs in general about what inclusive practice looks like and what that means for... yeah that individual child and family...”*
—Clinician

The role of clinicians in promoting and disseminating education on the benefits of PA for children with physical disabilities was identified by some clinicians as playing an important part in facilitating participation in football.


*“…if I said to them. ‘Do you wanna play footy?’ or ‘do you wanna play sport?’…and actually more actively encouraging either their parents or the child to think of it as an option and encouraging it for those reasons you know... you have your green prescription from the doctor... this will be the green prescription from your therapist... Find a sports team and go and play”.*


### 3.5. Attitudes and Expectations

Parental expectations relating to inclusion were frequently discussed as influencing whether parents enrolled their child in community-based football. Clinicians and parents reported that while some parents preferred their child to be in an adapted team or program, others preferred their child to play in a mainstream team.

A clinician reflected that parents’ worry about whether a child would be accepted and included may be a barrier to participation and lead to a consequent preference for individualised sports.


*“There’s a lot of worry for the child’s experience of rejection, or failure, or you know perhaps not failure but not success”*
—Clinician

A clinician identified parental apprehension about safety or the ability of child to participate as a barrier to participation, where football was described as being potentially dangerous or risky due to being fast-paced and involving complex skills like tackling and bouncing.

Club expectations of the child’s involvement in the game in terms of being performance-focused or enjoyment-focused was noted by a clinician as contributing to the experience of community football:
“I think having the culture of enjoyment without the competitiveness [is] the primary thing. So, I guess within the club culture, promoting participation as the main thing. Let’s still win, we don’t have to take scoring out of the equation, everyone knows who wins I guess a shift in that culture”—Clinician

Similarly, clubs’ attitudes towards children with disabilities as reflected by coaches, assistants and volunteers were also seen as a factor influencing participation. A parent of an eight-year-old girl reflected on the positive experience she had had when enrolling her daughter in a community football program.


*“…the coach was very welcoming. And so, you know, that’s fine, we don’t have an issue with children with disabilities coming and joining in and yeah”.*


A clinician also emphasised the importance of football clubs assessing coach/volunteer attitudes towards accepting children with disabilities before recruiting them.

Parents and clinicians also suggested that having coaches with disabilities involved in the session would act as a facilitator for participation for children with disabilities. One parent reflected on an experience her daughter had at a local basketball club:
“…her gymnastics coach…has CP and that was, she had a first session last week… walked away from that and said, ‘Oh that’s so good mum, she gets me, she understands me’. So, having that kind of mentor and role model who has a disability and is quite capable and has competed in umm like Paralympics type stuff has kind of given her inspiration, ‘yes I can do this’, and confidence”—Mother of an eight-year-old girl

### 3.6. Game Factors

The game requirements related to football were discussed as a barrier to participation for children with physical disabilities by both clinicians and parents. One clinician reflected on the game rules being challenging for children with additional needs.

Some clinicians suggested the use of adapted rules to facilitate inclusion for children with physical disabilities.


*“So what if the ball is kicked off and then they have... you know no-one’s allowed to move for three seconds and that child gets a bit of extra time...urm or if it’s thrown to them, you can’t tackle them or take the ball off them until you know three second, five seconds or whatever so that they have an opportunity to do something with it before it’s... they’ve lost the ball or yeah...”*
—Clinician

A clinician also noted that the speed of Australian Rules Football can often be too fast for children with CP. Furthermore, it was noted, due to the varying cognitive capacities, that children may not understand some of the common commands used in football such as *“stop, free pass…”.*

The use of aids and tools to accommodate physical challenges was discussed as a facilitator to participation.

Some clinicians also discussed the use of visual aids to help children understand verbal commands:
“... and for some kids, having a [repeated] picture to represent a word or to represent communication can be so incredibly valuable, especially in a busy, chaotic, distracting environment where the speech comes and goes and its gone, but if an umpire or a coach could hold up a symbol that said what the call was, or what the thing they’re doing was, just something very simple and clear. You know umpires could wear a lanyard that had 10 symbols...”—Clinician

The timing of the football season was also seen as a barrier in relation to it being a winter sport and the environment often being subject to wet weather. The high contact nature of the sport was also identified as being barrier to participation.

Clinicians also highlighted that further barriers in the physical environment, particularly that the size of the ground, may make participation challenging for children with physical disabilities, with one mentioning it’s a *“huge ground”* and another commenting that *“the surface might be an environmental factor as well”.* The shape of the football was also seen as potentially challenging for children with physical difficulties to manage.

Flexibility with locations and ground/surfaces used was discussed as facilitating participation. Clinicians discussed options such as using only half size grounds and moving sessions inside during wet weather:
“*If urm environmental issues became a problem but there was openness to consider adapting the environment, so whether it be the surfaces or the size of the ground or whatever comes urm comes with that might need to be considered that there was a willingness to consider other options within the sport”*—Clinician

Fatigue was also identified by clinicians as a barrier to participation—Clinicians highlighted the need to make benches and chairs available for children who became fatigued during the game.

The time of the week and day the sessions were held were also discussed as a barrier, where a parent of an eight-year-old girl mentioned that for her child, having sessions on a Friday night was challenging as by that point *“she’s exhausted”*. A father of a six-year-old boy also commented on session length being too long *“he [his son] won’t last an hour let alone an hour and a half”*.

A mother of an 8-year-old girl suggested that the session length be reduced to 45 min.

She also suggested that sessions be held earlier in the week:
“…if it was offered on a Monday or Tuesday night, earlier in the week before she is tired from the week at school”.

This parent also offered the suggestion of running intensive *“school holiday*” programs which would replace weekly sessions throughout winter.

An openness to diversifying the nature of participation to include children with additional needs was frequently identified as a facilitator to participation:
“Children with disabilities or young adults with disabilities tend to be given more of the role of a helper or run the bench or statistician or you’re an umpire or something like that. So, you’re still involved but with a different role typically…I think it’s becoming a little bit more open”—Clinician

A clinician also discussed that that pairing up children as part of a Buddy system would facilitate participation:
“I also like pairing up kids, I think that works nicely and having sort of a buddy system where everybody is, if you get the ball, you have to pass to your buddy before the next person goes…pair someone whose sort of better skilled than someone who is less skilled, so one of the people that gets the ball all the time then has to pass the ball to the person that doesn’t get the ball …”—Clinician

### 3.7. Community and Cultural Relevance

The presence of a community culture and sense of communal identity and belonging around sport and local sporting clubs was identified by both parents and clinicians as a major facilitator to participation in community football. The local football club was seen as a large part of the community beyond being sporting providers:
“One of the big standouts for me for any team or club is about that, you’re involved in the community, you’re part of something, there’s a relationship and there’s connections being made”—Clinician
“Families would go down and hang out at the footy and kick alongside the ground.”—Clinician
“The parents love the fact that they’re (the child) getting that sense of connectedness”—Clinician

The importance of belonging to a club for some children with physical disabilities was also highlighted by clinicians:
“I’ve got a number of kids that would love to be just part of a club, but maybe have their own team with altered rules and then but still you know still come to the club rooms after and still get best on ground and still do all that sort of stuff but it’s sort of an all-abilities group—Clinician

Clinicians emphasised the popularity of football within the community:
“I think the love of football is very strong in Geelong…”—Clinician

Parents also commented on the importance of football in not just the specific community, but the wider state.

Both parents and clinicians commented children also sometimes discussed football in their school settings, for instance, which teams they supported and players on those teams. One clinician also indicated that many children they see seem to have an interest in football teams however do not have an interest in playing the sport.

## 4. Discussion

The current study adopted a qualitative approach to explore parent and clinician perspectives on the factors influencing participation in community-based Australian Rules Football for primary school-aged children with CP in a regional setting. Clinician perspectives have been suggested to be a valuable source of knowledge on the facilitators and barriers to participation in PA [32]. In addition to child-related factors, which were predominantly non-modifiable, a range of environmental factors were identified by clinicians and parents as influencing participation.

Clinicians and parents largely discussed similar personal factors influencing participation—many of which aligned with existing empirical findings. While the parents in this study were of children with independent mobility, clinicians provided perspectives on the participation of children who required assistance with mobility. Aligning with personal factors identified within the concept of Body Structure and Function in the ICF framework [13], the child’s cognitive and physical abilities were discussed as a potential barrier to participation. Clinicians’ and parents’ reflections on the heterogeneity in the type and severity of physical, cognitive, sensory and communicative impairments in CP populations highlighted the importance of tailoring activities to suit this broad range of abilities. Aligning with this finding, the existing literature suggests that energy levels, fatigue and pain are areas of concern for parents of children with CP when participating in PA [37]. An interest in football was seen to be a core factor in facilitating the child’s decision to enrol in the activity, aligning with the notion of preference outlined in the Family of Participation Related Constructs (fPRC) [12]. It was suggested that as the child moved from early to middle childhood, their interest in trying a new activity such as football may increase, but that this interest in participation may decrease as they progressed through primary school, a trend reflected in current research exploring participation in children with physical disabilities [38]. Furthermore, a child’s confidence in their skill level was perceived to be a key facilitator to participation, where children with greater insight into their impairments were thought to compare themselves to their peers and be more likely to lack confidence in participation. Given that confidence is a central aspect of sporting performance, as suggested by previous research [39], there is a need for personalised PA interventions which address the child’s level of confidence, alongside a consideration of the factors influencing their confidence. With the exception of confidence, the personal factors discussed by both parents and clinicians were largely non-modifiable and implicated the major role of the environment in accommodating these factors in order to facilitate participation in football.

Clinicians identified the availability of parental and club resources as an important factor influencing participation. Consistent with findings of time and financial strain associated with caring for a child with a disability [40], clinicians reflected that parents of children with CP may have limited capacity to commit to PA programs for their children due to competing demands on their family’s time and finances, for instance, medical and therapy appointments. Furthermore, the physical resources of the club enabling accessibility was identified as a facilitator for participation, where accessibility plays a pivotal role in facilitating participation [41]. Specifically, the club’s ability to modify the physical environment in aspects not directly related to the game (e.g., car parking, toilet access) was seen to be important. Accessibility has suggested to be a significant factor accounting for variations in levels of participation between geographical regions [27,28]. Information being available to families about the accessibility of the club was recommended as a facilitator for involvement via a nominated contact person. Secondly, the club’s capacity to provide opportunities for the child to try out football before enrolling in the activity was discussed to be helpful in encouraging the child to decide if they would enjoy the activity. Indeed, theories of graded exposure suggest that gradual exposure to a potentially stressful event can reduce anxiety leading to the event [42].

Consistent communication between coaches, parents and clinicians were highlighted by both parents and clinicians as imperative to facilitating the child’s participation in community football, via the facilitation of knowledge-sharing. Specifically, a collaborative approach to the child’s participation was suggested, where the parent, coach and clinician had a centralised means of communication where they were able to share knowledge about the child as well as challenges and strategies to implement to facilitate participation. The collaborative approach suggested encapsulates elements of case management models used in clinical and community settings [43], where families and service providers work together to develop goals and interventions to optimise outcomes based on the needs of the child. The discussions arising from this study suggest that a coordinated ‘care-team’ approach in relation to the child’s participation may facilitate participation by enabling the development of tailored and appropriate PA interventions for children with cerebral palsy. Within the context of this approach, both parents and clinicians agreed that the parent played a vital role in advocating for the child and their needs with coaches and clinicians to facilitate inclusion, mirroring the current literature indicating that parent advocacy promotes equity in special needs educational settings [31] and in community settings [27].

Aligning with previous findings, the current study indicated that prior exposure to football via parental and/or sibling involvement in PA was a facilitator for participation [14,22]. This further compounds the learned behaviour and modelling elements of participation proposed in the systematic review of PA participation in CP populations by Shields, Synnot and Barr [14]. Clinicians also identified that providing families and children with information on football as an option for an activity, alongside its benefits for them may encourage the child to participate. Similarly, Wright, Roberts, Bowman and Crettenden [22] also found that young people with CP were more likely to participate if they were aware of the benefits of PA participation, highlighting the important role of education and knowledge-building in supporting the sustained participation of children with CP in PA. Brunton [32] suggests that clinicians are well-placed to inform and educate children and families about the benefits of PA and options for participation in their community.

Clinicians also suggested that a lack of experience and skills in coaching children with a disability was a key barrier to creating an inclusive environment in community football programs. Specifically, it was suggested that not having knowledge about specific conditions and the resulting risk of injury may adversely impact on coaches’ confidence and consequently their willingness to coach children with disabilities. Staff skill level [14] and support from staff in the form of assistance and information [27] have been found to be a key factor influencing overall activity participation in children and youth with CP. It was suggested in the current study that training for coaches around inclusive practices would be important in addressing the barrier of a lack of knowledge and confidence around coaching children with disabilities.

Attitudes and expectations of both parents and coaches were seen to influence participation in community football for children with CP. Parental worry around the child’s experience of failure or rejection by peers and coaches, as well as worry about physical injury were both discussed as barriers to participation. Aligning with the current findings, previous studies have also identified the suitability of physical activities as a key concern for parents of children with CP when making decisions about participation [37]. Previous positive experiences of inclusion of other children in a particular club, for instance, or having an older child enrolled in the club increased the likelihood of parents enrolling their child in the club. Parental preferences were varied in relation to whether parents preferred their child to play in a mainstream team with typically-developing peers, or in an adapted team consisting only of children with disabilities. The range of preferences once again highlights the importance of offering a variety of participation options via tailored programs in order to meet the needs of a range of families, rather than a one-size-fits all approach, as proposed by Morris, Imms, Kerr and Adair [24]. Coaches, assistants and volunteers who demonstrate an inclusive and welcoming attitude towards children with disabilities were seen to facilitate participation, a finding consistently reflected in the literature examining facilitators and barriers to activity participation in children with CP broadly [14,24]. Promoting a culture of enjoyment over a performance-driven culture was found to be another facilitator for participation. Theoretical frameworks such as the F-words framework [44], as well as empirical findings have identified fun and enjoyment as a salient facilitator of participation in children [5]. The involvement of coaches and athletes with a disability were seen to facilitate participation by increasing the child’s level of motivation and confidence, a finding also indicated in the study by Lauruschkus, Nordmark and Hallstrom [37].

The nature of Australian-rules football was found to pose a range of barriers to participation for children with CP, including the game moving too fast, the potential complexity of commands for children of varying cognitive abilities (e.g., stop, free pass), the large size of the ground and the shape of the football, which was seen to be difficult for some children with physical disabilities to manage. The timing and length of the sessions were also suggested to be potential barriers, where parents suggested that 60–90 min was too long, and their child would often get fatigued. Sessions earlier in the week were seen to facilitate participation, as the child’s energy levels would be higher than after a busy week at school. Activity limitations occurring in the context of physical and biological factors are indeed a significant barrier to participation in populations of children and adolescents with CP and physical disabilities more broadly [13,22]. Parents and clinicians in the current study proposed a range of environmental modifications to facilitate participation, including adapting the rules of the game to enable greater involvement for children with CP, introducing buddy systems, diversifying the nature of participation so children with CP had options to select different roles (e.g., umpiring), and modifications to the physical environment in which the game was played, including playing on half-size grounds, playing the game indoors on rainy days, introducing aids such as cones, and ensuring benches and chairs were available for children to rest. Given the heterogeneity of physical, cognitive, sensory and communicative impairments seen in CP, the ‘care-team’ approach with input from the child, parents, clinicians and coaches will enable a comprehensive consideration of the child’s unique needs and preferences when considering participation in community-based PA.

The social aspect of participating in community football was found to be a salient facilitator for participation. Echoing many previous studies exploring participation in youth with CP [37,45], both parents and clinicians agreed that the sense of belonging and connection to peers and to the wider community facilitated participation in organised sport for children with CP. Being a community-based activity, this sense of belonging extended beyond playing the sport, to peer relationships on and off the field, for instance, discussing football teams with peers at school. The finding that football is an ingrained aspect of belonging within the regional community in this study further builds on the conceptualisation of Environmental factors within the ICF framework, and highlights the potential need to take into account broader factors relating to community identity and belonging when considering the factors influencing participation in community-based PA in regional settings. Future research exploring the community connectedness in community-based PA in metropolitan settings will shed light on the relevance of this concept as a facilitator in community sport outside of a regional setting.

### 4.1. Limitations

There were a number of limitations to this study. Firstly, given the relatively small sample size of eight, findings from the current study should be interpreted with caution and further consolidated in larger sample sizes. Despite advertisement of the study to parents in a range of settings within the regional community as described above, over a 10-month period, only two parents volunteered to participate in this study. The low rate of engagement with the study is likely reflective of the significantly lower rates of participation and engagement in PA seen in children with CP when compared to typically-developing children. Another challenge to recruitment relates to the current focus on a regional setting, whereby this pool of families of children with CP and clinicians working with children with CP was significantly smaller than in metropolitan areas. The small number of parents recruited in this study further highlights the need to understand and address the barriers to community-based PA participation faced by children with CP and their families in regional settings.

In a review of qualitative health research over a 15-year period, Vasileiou, Barnett, Thorpe and Young [46] recommend that sample size should be considered in relation to the adequacy of the data collected, with reference to the specific features of the study itself. While limited enrolment into the study hindered the achievement of data saturation, the focus groups nevertheless elicited rich data aligning with current theoretical and empirical findings of factors influencing participation in children with disabilities, which should be further built on in larger-scale qualitative studies.

Secondly, while the perspectives of parents and clinicians provided a multi-layered view of the facilitators and barriers to participation, focus groups were not conducted with children. Future research comparing and contrasting the themes derived in this paper with those of children with CP, as in studies such as those of Wright, Roberts, Bowman and Crettenden [22], will provide a more holistic view of the facilitators and barriers to participation in young people with CP.

Lastly, given that the parents in this study were of children with independent mobility, it would be important for future studies to capture the perspectives of parents of children who required assistance with mobility.

### 4.2. The Need for A Holistic Approach

The facilitators and barriers identified in the current study strongly echo existing theoretical and empirical research on the factors influencing participation in organised PA. The current study suggests that with the exception of factors in the physical environment, the environmental facilitators and barriers to participation in PA for children with CP are influenced largely by ‘people factors’, specifically, the roles of parents, clinicians and coaches within the broader club setting. Based on the themes arising from this study, Figure 2 provides a summary of recommended roles of the child’s support system in relation to facilitating community-based PA participation for children with CP.

Given the relatively low PA participation rates [8] and the known benefits community-based PA participation for children with CP [2,3], there is an urgent need for a framework of guiding principles to inform the development of tailored community-based PA interventions based on a ‘care-team’ approach, which takes into account input from the child, their parent, coaches and clinicians. Drawing on the key themes arising from this study, the following 3Cs (Communicate, Consider, Collaborate) model is proposed for the care-team approach facilitating participation:
**Communicate**: *For parents, clinicians and club staff to develop a system of communication that is appropriate for the whole system, prior to the child commencing participation (e.g., developing a shared summary document of the child’s preferences, strengths and weaknesses that can be modified throughout the season). The role of the parent within the care-team in advocating for the child is particularly relevant in this instance.*
**Consider**: *For each party within the care-team to engage in self-reflection about their individual contributions to the system in facilitating community-based PA participation for the child. This entails reflection on strengths they can bring (e.g., expert knowledge on a sport or disability, mentoring players with disability as a coach with a disability), as well as any challenges they may anticipate with regards to facilitating participation for the child and how they can work with the care-team to problem-solve this (e.g., attitudes towards disability, or the physical club facilities not being wheelchair accessible).*
**Collaborate**: *For parents, clinicians and club staff to develop opportunities for knowledge-sharing and joint decision-making on tailored adaptations for inclusive practices (e.g., tip-sheets by clinicians with disability-specific information, introducing a buddy system). The clinician’s provision of disability-specific information and how it may impact on the child’s participation, alongside the club’s input on centre specific information (e.g., club expectations, facilities), will facilitate the development of tailored interventions based on the child’s needs and preferences and the consistent use of strategies across contexts.*

## 5. Conclusions

The current study highlighted the importance of the system around the child in facilitating PA participation within their community, specifically parents, clinicians and coaches. Clinicians were found to have a unique role facilitating participation in community-based PA, despite a current lack of focus on this group in the participation literature. A consideration of the roles of the child’s surrounding system collectively rather than in isolation, alongside approaches which enable communication and collaboration will help bridge the current gap in the holistic consideration of how this system can aid in the facilitation of PA participation, in the context of the child’s specific characteristics and preferences.

## Figures and Tables

**Figure 1 ijerph-17-01102-f001:**
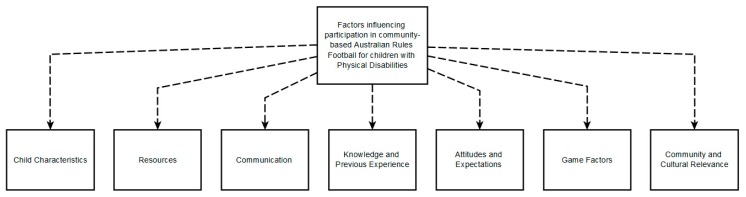
Themes arising from qualitative analyses relating to factors influencing participation in community-based Australian Rules Football for children with physical disabilities.

**Figure 2 ijerph-17-01102-f002:**
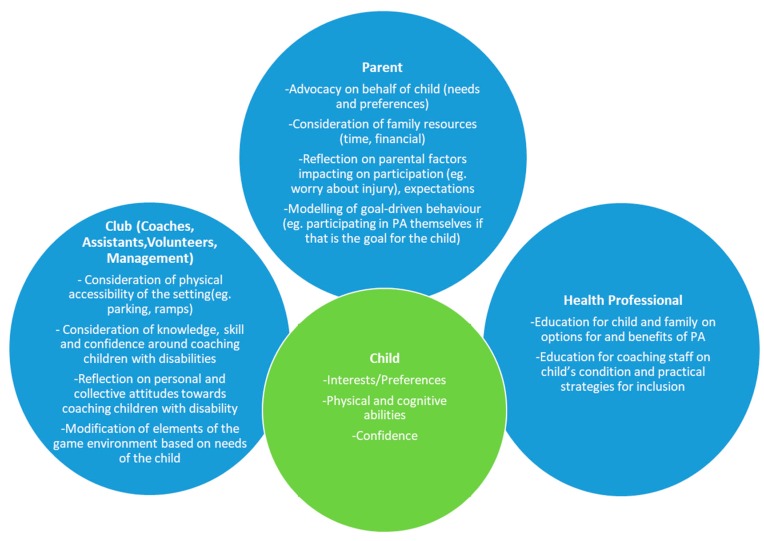
Proposed roles of parents, clinicians and the sporting club in facilitating community-based PA participation.

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
