# Peer review of "Parent and Clinician Perspectives on the Participation of Children with Cerebral Palsy in Community-Based Football: A Qualitative Exploration in a Regional Setting"

_ijerph, 2020, doi:10.3390/ijerph17031102_

Round 1

Reviewer 1 Report

Dear authors, thank you for the opportunity to review your manuscript titled "Parent and Clinician Perspectives on the Participation of Children with Cerebral Palsy in Community-Based Football: A Qualitative Exploration in a Regional Setting". This study makes a contribution to the literature on barriers and facilitators to physical activity participation in children with cerebral palsy (CP). I have some comments which must be addressed before this manuscript is suitable for publication, these are numbered below.

General/throughout

(1) The manuscript is very long. There is repetition and redundancy in the introduction and the discussion could be condensed. I would suggest searching for opportunities to condense the manuscript where possible. There are too many instances of this to comment individually.

(2) You define the abbreviation PA for physical activity in the first line of the manuscript however this acronym is not used consistently (e.g. 'physical activity' long form seen on line 33,34,37 etc.).

Introduction

(3) Line 43: the F-words framework is not related to the ICF however the way you have constructed this sentence makes it seem like they are. Consider revising.

(4) Line 44-47: This statement an understanding of environmental and social barriers is imperative in providing a holisitic conceptualisation of participation in PA cannot be attributed to the ICF and/or F-words. I would suggest using the 'Family of participation-related constructs' (Imms et al. 2017 DMCN) to discuss how environment/context and other constructs are related to one another and how they influence (and are influenced by) participation.

(5) Line 51: individually-based internal characteristics is repetitive, consider revising.

(6) Line 53: motivation is listed as a body function in the ICF however disposition is a personal factor and there is an overlap between what are body functions and personal factors in this area. It is probably not necessary in your introduction to make a distinction as it this background is not required for your study findings.

(7) Line 62: Congenital heart condititions (s4100) and obesity (b350) are characterised as impairments in body structures/functions in the ICF so are not needed to be listed here separately as medical conditions. Consider condensing this section.

(8) Line 68-82: This paragraph is structured awkwardly and does not adequately describe the gaps in the existing literature. There are quite a number of studies discussing barriers and facilitators to sports participation for children with CP including both quantitative and qualitative studies. What exact gaps are there that provide the scene for your study? i.e. no studies from regional areas, no studies of AFL, no studies contrasting parent and clinician views etc.

(9) Line 88 reference 29: two better references would be either Reedman S, Boyd RN, Sakzewski L. The efficacy of interventions to increase physical activity participation of children with cerebral palsy: a systematic review and meta-analysis. Dev Med Child Neurol. 2017;59(10):1011-8. or Bloemen M, Van Wely L, Mollema J, Dallmeijer A, de Groot J. Evidence for increasing physical activity in children with physical disabilities: a systematic review. Dev Med Child Neurol. 2017;59(10):1004-10. as these are specific to CP and physical activity

(10) Line 95-100: Functional therapy has a special meaning (i.e. goal-directed task-specific training) and does not mean here doing functional things in therapy (hopefully everyone is doing that!!). Revising this paragraph to talk more about facilitation of participation in therapy would be helpful. See Reedman SE, Boyd RN, Elliott C, Sakzewski L. ParticiPAte CP: a protocol of a randomised waitlist controlled trial of a motivational and behaviour change therapy intervention to increase physical activity through meaningful participation in children with cerebral palsy. BMJ Open. 2017;7(8). and Palisano RJ, Chiarello LA, King GA, Novak I, Stoner T, Fiss A. Participation-based therapy for children with physical disabilities. Disabil Rehabil. 2012;34(12):1041-52. for concepts. 

Materials and Methods

(11) You only have 6 clinicians and 2 parents (total of 8 participants and 3 focus groups). This is a very small sample size for a qualitative study whereby generally >10 participants is desirable and data saturation is rarely achieved with numbers less than this unless there has been a consistent experience. You must discuss this - why you have a small sample size, whether you reached saturation of themes, and what the impact of a small sample size could have been. Some of this belongs in the limitations section.

Discussion

(12) Lines 504-509: This is a long compound sentence, consider revising.

(13) Lines 513-514: This sentence is contradictory, consider revising.

(14) Lines 555-559: This is a long compound sentence, consider revising.

(15) Lines 576-577: This sentence doesn't make sense. Consider revising for clarity.

Reviewer 2 Report

This is a review of the manuscript entitled “Parent and Clinician Perspectives on the Participation of Children with Cerebral Palsy in Community-Based Football: A Qualitative Exploration in a Regional Setting.” A qualitative design was used to answer the research question related to the barriers and facilitators of participation of children with cerebral palsy (CP) in leisure involvement.

Overall, this paper elicits an interesting discussion regarding the perceptions of clinicians and parents into the active involvement of children with CP in leisure.  This is a timely paper, as findings highlight the complex nature of participation for children with CP, as well as many other important contributions to the field.  This manuscript was well written, and the flow of ideas were sequential.  To strengthen this manuscript, several prominent recommendations are listed below.

The introduction was well written and did a nice job of narrowing the focus to the research question.  It was the clearest of all sections in the manuscript.  My sole recommendation is to directly state your research question in your final paragraph, or, if you prefer, at the beginning of your analysis. There was some ambiguity related to the question.  For instance, did your study measure barriers, facilitators of participation, or factors influencing participation.  These examples all illustrate slightly different meanings.  To improve cohesiveness, ensure that the question aligns with findings.

In the “Participants,” section on pg. 3, the authors should consider providing a more detailed explanation of why various clinicians (i.e., physiotherapist, speech therapist, occupational therapist), were selected for this study, as opposed to a single type of clinician or other types of therapists.  Additionally, parent focus group participants were limited (i.e., two participants), and is a limitation of this study that should be addressed, since it is not typically viewed as an adequate sample size.  Questions about why additional parents weren’t recruited raises concerns about credibility of results. 

In regard to data analysis (pg. 4), there are several suggestions for strengthening this manuscript.  First, the authors did not discuss if additional data were analyzed beyond transcriptions (e.g., data that were potentially collected referenced on lines 135-136).  If these data were not included, it should explicitly be stated.  Second, and more importantly, measures taken regarding trustworthiness and credibility were limited. Given the qualitative nature of this work, if some or none of these measures were completed, it will be imperative that the authors are overt about these limitations.  For instance, piloting of the interview questions as well as member checking were never disclosed.  To strengthen analysis, it is critical that the authors specifically address (a) how the interview questions were derived; (b) the piloting procedures in place to ensure credibility of questions; (c) first and second level member checking procedures with participants; and/or (d) any other additional procedures leading to data trustworthiness. 

In general, results were interesting (pgs. 4-14), and add to the depth of our knowledge in this area.  However, revisions to this section will help to provide greater clarity.  The themes appear to be underdeveloped and could benefit from additional analysis (e.g., collapsing of several ideas into 3-5 very cohesive themes that answer the research question).  The over-use of quotes was distracting, and I recommend using a handful of the most salient and meaningful quotes. 

Table 1 (pgs. 12-14) was rather interesting with several unique strategies, but seems more appropriate for a practitioner manuscript, and did not clearly link to the research question.  Again, clarification of the research question might help in this regard, to create a more cohesive link from the table to the study findings.  Figure 2 (pg. 18) appears to be in more alignment with your study.  However, improvements made to the graphic design of this figure are recommended.

Like the introduction, the discussion section (pgs. 14-17) was well-written and connected closely to the introduction in addition to research findings. 

While the manuscript offers interesting insights into the perceptions of clinicians and parents of children with CP, the manuscript would benefit from improved clarification, expanded analysis, and cohesiveness in specific sections.

Round 2

Reviewer 1 Report

Dear authors,

Thank you for the considerable revisions you have made to the manuscript addressing reviewer queries. The manuscript has been strengthened by these revisions.